# A Narrative Review on the Collection and Use of Electronic Patient-Reported Outcomes in Cancer Survivorship Care with Emphasis on Symptom Monitoring

Corina J. G. van den Hurk [1,*], Floortje Mols [1,2], Manuela Eicher [3,4], Raymond J. Chan [5], Annemarie Becker [6], Gijs Geleijnse [1], Iris Walraven [7], Annemarie Coolbrandt [8,9], Maryam Lustberg [10,11], Galina Velikova [12], Andreas Charalambous [13,14], Bogda Koczwara [15], Doris Howell [16], Ethan M. Basch [17] and Lonneke V. van de Poll-Franse [1,2,18]

1 Department of Research and Development, Netherlands Comprehensive Cancer Organization (IKNL), 3511 DT Utrecht, The Netherlands; f.mols@tilburguniversity.edu (F.M.); g.geleijnse@iknl.nl (G.G.); l.vandepoll@iknl.nl (L.V.v.d.P.-F.)

2 CoRPS—Center of Research on Psychological Disorders and Somatic Diseases, Department of Medical and Clinical Psychology, Tilburg University, 5037 AB Tilburg, The Netherlands

3 Institute of Higher Education and Research in Health Care (IUFRS), Faculty of Biology and Medicine, University of Lausanne and Lausanne University Hospital, CH-1010 Lausanne, Switzerland; manuela.eicher@chuv.ch

4 Department of Oncology, Lausanne University Hospital, CH-1011 Lausanne, Switzerland

5 Caring Futures Institute, College of Nursing and Health Sciences, Flinders University, Adelaide, SA 5042, Australia; raymond.chan@flinders.edu.au

6 Amsterdam UMC, Department of Pulmonary Diseases, Cancer Center Amsterdam, 1081 HV Amsterdam, The Netherlands; a.becker@amsterdamumc.nl

7 Radboudumc, Department for Health Evidence, 6525 GA Nijmegen, The Netherlands; iris.walraven@radboudumc.nl

8 Department of Oncology Nursing, University Hospitals Leuven, 3000 Leuven, Belgium; annemarie.coolbrandt@uzleuven.be

9 Department of Public Health and Primary Care, Academic Center for Nursing and Midwifery, 3000 Leuven, Belgium

10 Breast Medical Oncology, Yale Cancer Center, Yale School of Medicine, New Haven, CT 06520, USA; maryam.lustberg@yale.edu

11 Breast Center at Smilow Cancer Hospital, New Haven, CT 06519, USA

12 Leeds Institute of Medical Research at St James's, University of Leeds and Leeds Cancer Centre, St James's University Hospital, Leeds LS9 7TF, UK; g.velikova@leeds.ac.uk

13 Nursing Department, Cyprus University of Technology, Limassol 3036, Cyprus; andreas.charalambous@cut.ac.cy

14 Department of Nursing Science, University of Turku, 00074 CGI Turku, Finland

15 Flinders Medical Centre, Flinders University, Adelaide, SA 5042, Australia; bogda.koczwara@flinders.edu.au

16 Princess Margaret Cancer Research Institute, University of Toronto, Toronto, ON M5G 2M9, Canada; doris.howell@uhn.ca

17 Lineberger Comprehensive Cancer Center, University of North Carolina Cancer Center, Chapel Hill, NC 27599, USA; ebasch@med.unc.edu

18 Department of Psychosocial Research, Division of Psychosocial Research & Epidemiology, The Netherlands Cancer Institute, 1066 CX Amsterdam, The Netherlands

* Correspondence: c.vandenhurk@iknl.nl

**Abstract:** Electronic patient-reported outcome (ePRO) applications promise great added value for improving symptom management and health-related quality of life. The aim of this narrative review is to describe the collection and use of ePROs for cancer survivorship care, with an emphasis on ePRO-symptom monitoring. It offers many different perspectives from research settings, while current implementation in routine care is ongoing. ePRO collection optimizes survivorship care by providing insight into the patients' well-being and prioritizing their unmet needs during the whole trajectory from diagnosis to end-of-life. ePRO-symptom monitoring can contribute to timely health risk detection and subsequently allow earlier intervention. Detection is optimized by automatically generated alerts that vary from simple to complex and multilayered. Using ePRO-symptoms during in-hospital consultation enhances the patients' conversation with the health care provider before

making informed decisions about treatments, other interventions, or self-management. ePRO(-symptoms) entail specific implementation issues and complementary ethics considerations. The latter is due to privacy concerns, digital divide, and scarcity of adequately representative data for particular groups of patients.

**Keywords:** cancer; survivorship; electronic patient-reported outcomes; symptoms; eHealth; quality of care; quality of life; self-management; ethics

## 1. Introduction

This narrative review aims to describe the collection and use of electronic patient-reported outcomes (ePROs) for cancer survivorship care, with an emphasis on ePRO-symptom monitoring. Worldwide, ePRO data collection is now gradually being implemented in survivorship care [1,2], and related literature is still limited. We therefore mainly present results from research settings. We used the definition of survivorship as the process that begins at the moment of diagnosis and continues until end-of-life [3].

The paper is subdivided in three sections: First, we introduce the methodology and implementation of ePROs in general. Second, we describe ePRO-symptom monitoring including outcomes, methodology, alerting algorithms, symptom self-management, and implementation. Third, we discuss ethical issues followed by concluding remarks. The narrative methodology is used to obtain a broad perspective. We did not intent to locate all relevant literature in a systematic search.

## 2. ePRO Methodology and Implementation

### 2.1. ePRO Methodology

PROs are defined by the Food and Drug Administration (FDA) as: ' . . . any report of the status of a patient's health condition that comes directly from the patient, without interpretation of the patient's response by a clinician or anyone else' [4]. Examples of PROs are symptoms, health-related quality of life (HRQOL) (with domains such as physical, psychological, and social functioning), self-efficacy, anxiety or depression. An electronic PRO Measure (ePROM) is an online question or questionnaire that is used to collect ePROs.

ePROMs are completed online or via a (telephone-based) interactive voice response system [5]. ePRO data collection is labeled to be active when the patient self-reports and passive when using devices such as wearables and monitoring tools on smartphones (e.g., sleep registration applications). Both methods are considered to be ePROs and help provide information of the patients' health and disease. The emphasis in this paper will be on active ePRO reporting.

#### 2.1.1. Selecting Questionnaires and Providing Feedback

A very large number of generic and cancer-specific ePROMs are available. For research purposes, e.g., the Dutch 'Patient-Reported Outcomes Following Initial treatment and Long-term Evaluation of Survivorship' (PROFILES) registry incorporates 60+ ePROMs for cancer patients [6,7]. For use in routine cancer care, guidance in choices for ePROMs was recently published by Cancer Care Ontario. They stated that selection should incorporate topic coverage, usability, and psychometric properties [8]. Topic coverage means the relevance for most cancer patients, cancer type, or cancer treatment and according to the phase of the disease (diagnosis, during or following active treatment) [9]. For example, patients receiving immunotherapy might not be able to adequately report their symptoms and HRQOL when using ePROMs developed in the era of chemo- and radiotherapy [10]. Usability includes conceptual characteristics (e.g., type of scale, and recall time frame), scoring (e.g., domain, item, and/or global scores), time to complete the measure, use of plain language, and licensing or fees for use. Psychometric core considerations include internal

consistency, test re-test reliability, responsiveness, discrimination ability, meaningful change, and translation validity.

For selecting PROMs, the Patient-Reported Outcomes Measurement Information System (PROMIS) of the National Institutes of Health has been set up [11]. PROMIS uses item banks, short forms, and computer-adaptive testing to measure health-related constructs with fewer items than conventional questionnaires. A recent review summarized several PRO data collection methods with PROMIS, which is valid and reliable for use in the cancer population [12].

Definition of (e)PROM item sets for the emerging Value-Based Health Care (VBHC) [13] has been proposed for different cancer patient populations by the International Consortium for Health Outcomes Measurement (ICHOM) [14]. These standard sets are also meant to be used in routine cancer care and are published per cancer type. They have been developed by consortia of experts and patient representatives in the field and are combined with clinical items.

After completing ePROMs, patients and health care practitioners (HCPs) might receive (visualized) feedback. Feedback can be personalized comparing individual results with (1) their previous results; (2) with patients similar to them; or (3) with a normative population [15–17]. By being able to compare their own scores with a person of the same age and sex without cancer, a patient might be able to better interpret his/her own functioning. A recent review shows that PRO feedback results in improvements in cancer- and symptom control, HRQOL, communication, and patient satisfaction [18].

### 2.1.2. Timing

Longitudinal ePRO data collection is most often used; what is important to the patient in daily life related to cancer care can change over time. It might change during the survivorship trajectory from diagnosis until follow-up or in the transition from curative to palliative and end-of-life care [9,19].

In the longitudinal data collection, specific timings or frequencies for PROMS are chosen, depending on the purpose of ePRO data collection and the patient's preferences. During active treatment, ePRO-symptom collection has been reported to be feasible and acceptable at every clinic visit [20], weekly [21,22] or even daily [23,24]. In this phase, the purpose is to prevent and rapidly detect worsening of symptoms primarily in the light of health risks and secondarily to improve HRQOL. Relatively long-time intervals are used to monitor HRQOL or symptoms in the phase of follow-up compared to active treatment. In follow-up, the aim is to gain insight into how the patient is doing, to detect recurrences, or to identify (non-urgent) issues that might be mitigated or resolved at the next in-hospital consultation or through self-management interventions. If patients are able to manage mild or moderate symptoms themselves at home, they contribute to the prevention of worsening of symptoms and reduce hospital and General Practitioner (GP) contacts [25].

Another timing topic is to visualize in detail a patient's problem over the course of the day to discover patterns as a basis for intervention. In this so-called Ecological Momentary Assessment, the patient reports on his or her problems several times a day for a few weeks [26,27]. This is, for example, performed by automatic timing-based triggering following persistent high scores on an ePRO-symptom such as pain or fatigue [28]. A patient who feels exhausted every afternoon may find out that he/she is doing too many activities early in the morning and learns to better dose energy throughout the day. A last example of outcome-based frequency determination is using ePROs to screen for financial toxicity. Such data collection may be less frequently undertaken, i.e., especially during key transition points [29].

### 2.1.3. Future Developments

In the future, ePROs will be routinely used in survivorship care. In order to provide personalized survivorship care, it is helpful if the patient and/or the HCP can decide what type of ePRO data collection and feedback suits the specific individual patient best. In an

ideal situation, choices can be made and activated online in the Electronic/Patient Health Record (EHR/PHR). These can be based on the patient's context and profile, detected unmet needs, and preferences, including the option of no ePROs.

In addition to providing feedback, there should be the ability to share ePROs intra- and extramurally with the multidisciplinary team, the GP, dietician or paramedics. This would add to the interdisciplinary coordination of care, especially in shifts and referral situations [2]. Information sharing enhances the value of the collected ePROs and it reduces the reporting burden on the patient. However, to establish this exchange, there are challenges such as implementation of local care pathways, but also general awareness of the added value of ePROs and the ability to interpret them [2].

### 2.2. ePROs Implementation in Routine Care

The use of ePROs for survivorship care influences the patient's involvement in monitoring and co-coordinating his/her disease, as well as the clinical workflow for HCPs. The added value of ePRO monitoring for both groups is highly dependent on the implementation. Implementation includes both adoption into routine cancer care as well as the strategies for this adoption.

Systematic reviews report limited adoption of ePROMs into routine care [30–32]. They describe common barriers such as technological and behavioral issues as well as how and why to use ePROs. In addition, context-specific enablers are identified such as easy to use technology, clear feedback of results, sufficient local resources, and long-term implementation support. Therefore, tailored implementation strategies need to be designed, based on the resources of each individual clinic [32]. Both the 'International Society for Quality of Life Research (ISOQOL) PROMs/Patient-Reported Experience Measures in Clinical Practice Implementation Science Work Group' as well as Cancer Care Ontario advocate the use of implementation science and frameworks [8,32]. Cancer Care Ontario has actually developed a framework, containing three key themes for implementation of PROMs: acceptability, outcomes, and sustainability [8]. Acceptability incorporates patient and HCP perspectives, as well as the clinic as a whole. Engaging all stakeholders and a clinic protagonist for each participating hospital is key for successful implementation [28,33]. Outcomes are about identifying and using the relevant ePROMs to improve care and add value. ePROMs are sustainable if they have the potential to be embedded in routine cancer care, including support and resources [8].

The primary stakeholders to be involved in implementation are patients, who have been found to be more willing to complete ePROs if the questions are relevant to them personally and if the results are actually used during consultation [2,21]. In addition, the ePROMs need to be easy to understand and easy to complete [8]. A review showed that patients generally liked to be able to complete questionnaires at home, accepted the (often extensive) length of the questionnaires, and were as well willing to answer additional questions [31]. Patients are better prepared and more willing to participate if they are offered to discuss the ePROMs process with an (research/nursing/lay) assistant who can spend more time addressing patients' questions than a clinician or nurse during in-hospital consultation [33,34]. Instruction videos about interpretation of scores and the use of the reports for communication with HCPs also foster patient activation [28,35]. Both should be incorporated in the clinical or study resources when planning ePROM data collection.

Further, HCPs are important stakeholders for implementation. They need to feel that the results are clinically relevant and outcomes amenable to change; they prefer ePRO collection that allows them to do something in response to a burdensome symptom or a worsening ePRO score [8]. They must be able to interpret the ePRO scores, understand the hows and whys, and integrate this in the clinical flow [8,32]. It is well embedded in the workflow if the ePRO collection is not considered an additional task but is rather a workload reallocation. In addition, it needs to be perceived that it meets a patient's need, reduces (undesirable) care variation, and optimizes the use of resources and multidisciplinary

care [1]. In the clinic, managers, information technology specialists, data managers, and security officers also need to be involved in implementation.

Future Developments

A recent review found that in most institutions, both patients and HCPs do not yet receive ePRO feedback reports in the stand-alone applications or in the EHRs/PHRs or dashboards [18]. Integration in the EHR/PHR is time consuming and costly and must be arranged separately in each individual institution. There are several levels of technical solutions to provide feedback. A multi-step approach can be followed, from simply pushing (static) feedback to the EHR, to single sign-on integration, to sophisticated deep and dynamic integration. These structural aspects are of utmost importance for the use of the results in the consultation room and therefore for successful implementation. This should be addressed at the outset, ideally within a health care system [36].

## 3. ePRO-Symptom Monitoring

In this review, symptom monitoring is defined as the process in which patients actively report symptoms through electronic devices at home (or in the hospital). It may also include feedback to the patient and/or HCP.

### 3.1. ePRO-Symptom Monitoring Outcomes

The overall advantages of symptom monitoring for quality of care and HRQOL in research settings are well known: Due to early intervention following alerts, patients experience less severe symptoms that prolong for a shorter period. In addition, they have survival benefits and a better HRQOL, they are less anxious, have an improved physical well-being, adhere longer to therapy, need fewer visits to the emergency room, are less frequently admitted to the hospital, and it increases their confidence in self-management [13,28,37–39]. In addition, ePRO symptom monitoring improves communication between the patient and HCP. As a result, it has the potential to enhance access to care as well as patient engagement, patient empowerment, and self-efficacy to manage symptoms [22].

Another advantage of monitoring and feedback is that the patient and HCP have an overview of the course of symptoms over time [21,25,33]. Therefore, trends of deterioration or improvement can be easily observed as well as the symptoms that require attention during in-hospital consultation. In contrast to what may be expected, clinicians report that feedback from symptom monitoring does not lengthen these consultations [40].

Future Developments

Amongst other advantages for the individual patient, symptom monitoring is expected to increase quality of care and patient safety by delivering a more correct identification of symptom gradings [41]. Firstly, in clinical trials, symptom reporting by clinicians usually focusses on grade 3 and 4 toxicity. In contrast, with ePRO-symptom assessment, the whole spectrum of grade 1 to 4 is collected, enabling more fine-grained and patient-centered toxicity and severity evaluation. Secondly, patients report different symptom profiles than the HCPs as patients tend to report earlier, more frequent, and higher symptom grades [42,43]. Symptoms remain undetected and unaddressed by HCPs up to half of the time [44,45]. Correctly diagnosing the severity of symptoms as experienced by the patient is important; it determines the intervention that will be chosen according to clinical symptom management guidelines, informs dose modification of cancer therapies, and facilitates patient-centered care.

Another advantage arises when ePRO-symptoms are also measured at baseline, which is uncommon in clinical trials and is rarely structurally recorded in routine care. Studies have shown that at baseline, many low- and high-grade symptoms are already present [41]. Therefore, these should not be attributed to the treatment, but to the disease and/or co-morbidities, and/or other patient characteristics and may require a different (timing of) interventions.

As ePRO-symptom monitoring in survivorship care is relatively new, its value for the prevention of persistent adverse events (AEs), i.e., long-term and late effects, is still unknown. Since many patients continue to suffer from persistent symptoms for many years after diagnosis [46–48], we expect early prevention through monitoring during treatment to be also beneficial for long-term HRQOL. This emphasizes the value of ePRO reporting across the entire survivorship trajectory from diagnosis until follow-up.

### 3.2. ePRO-Symptom Monitoring Methodology

Worldwide, many ePRO-symptom monitoring applications for cancer patients have been developed and tested in studies [5,49,50]. A recent rapid review showed mostly randomized trials and feasibility studies from Europe and North America to report these applications [5]. They had different content but contained essentially similar features and served similar purposes. Most applications sent alerts when a symptom exceeded a predefined threshold and provided feedback to the patient and HCP. The applications were intended for patients with different types of cancer and treatments. Some applications were enhanced with self-management advice and patient education or an eConsult function, or with additional triage questions. Triage automatically distinguishes between low-grade symptoms that only require reassurance or self-management advice and the higher-grade symptoms that need to be followed up by a HCP [5]. Most symptom monitoring infrastructures also captured other ePROs, such as HRQOL.

ePRO symptom monitoring has been shown to be feasible and can be of added value for patients of all ages. Most studies are performed among adults (30–90 years) [21,22,33,37–39,51], but it also has been initiated in adolescents and young adults [51] and pediatric patients [52]. The extent of usage is influenced not only by age but also by gender, race or education, as well as by the attitude of the HCP towards monitoring and the use of the results [21,22].

In the literature, various measures are used for ePRO-symptom monitoring. The most common are the Patient-Reported Common Terminology Criteria for Adverse Events (PRO-CTCAE), the Edmonton Symptom Assessment Scale/System (ESAS), and the Memorial Symptom Assessment Scale (MSAS) [5,49]. The ESAS represents one of the first symptom batteries, originated for usage in palliative care in 1991 [53]. Since then it has been extensively validated and translated into more than 20 languages. It has been implemented by Cancer Care Ontario where monthly more than 30,000–40,000 patients provide their ePRO-symptoms using ESAS [39].

The PRO-CTCAE [54] is also translated and validated in many languages worldwide and is mapped with the CTCAE [55]. This means that PRO symptom scores can be converted to toxicity grades of the CTCAE, which in turn allows us to determine the necessary intervention according to clinical guidelines.

### 3.2.1. Alert Algorithms for ePRO-Symptom Monitoring

Algorithms trigger alerts when symptom severity reaches a predetermined threshold. Alerts are sent to patients and/or to HCPs to initiate or intensify symptom management. These alerts enhance the usefulness and benefits of real-world ePRO-symptom data [49]. Monitoring algorithms are defined as any method that transforms data into prognostics or alerting systems. Alerting algorithms for symptom monitoring are diverse and range from simple to complex, multilayered systems that can include automated self-management advice [5]. Simple "reasoning algorithms" that trigger an alert based on a single parameter threshold (e.g., moderate or severe symptoms) and/or worsening of a symptom(s) are common in cancer symptom monitoring [37,56,57]. Similarly, applications and (telephone-based) interactive voice response systems set a conservative threshold of "quite a bit" or "very much" or a two-point worsening from the previous week to denote a clinically significant level of symptom severity for triggering of alerts [35,58,59]. These systems may also generate a single alert for multiple symptoms simultaneously as a cluster of symptoms.

Complex, multilayered alerting algorithms with multiple pathways have also been tested in cancer patients. For example, the Advanced Symptom Monitoring and Man-

agement System (ASyMS) was tested in several trials [60], and an adapted version uses reasoning algorithms based on more than one parameter in a risk-based scoring system [13]. In this system, alerts are graded (red and amber) based on a combination of symptom severity and other parameters (e.g., severe diarrhea combined with severe pain). The color defines the expected HCP response time: urgent red alerts have a 30-min time window to be addressed by the HCP, whereas amber alerts have an 8-h window. In the Canadian adapted version of ASyMS, parameters were further customized with additional questions. These questions are triggered for patient completion based on evidence-informed symptom triage guidelines (i.e., symptoms of dehydration plus moderate or severe diarrhea) to generate red alert risk levels [61]. The e-RAPID system for systemic therapy [62] and the system under development for post-surgical monitoring [63] stratify symptoms into three levels of symptom severity. For systemic therapy Level 1 is a mild or moderate symptom and the patient receives self-management advice. Level 2 means severe improving symptoms or a combination of moderate symptoms, and Level 3 is a medically severe symptom. In the algorithm, HCP alerts are generated for Level 3, and patients receive automated text messaging to contact the hospital for Levels 2 and 3.

### 3.2.2. Future Developments

In the future, ePRO-symptom monitoring applications can be made more patient-centered and relevant by adding a dynamic list of symptoms to be reported during specific treatment phases such as surgery, radiotherapy, or systemic therapy [7,64–66]. For the latter, symptom incidences vary highly between, for instance, chemotherapy and immunotherapy. Therefore, increased relevance would be achieved if the most anticipated symptoms for a particular (combination of) agent(s) are retrieved from a pharmacotherapeutic database and incorporated in the application. Giving patients the option to report symptoms that are not in the standard list also increases relevance for the patient.

### 3.3. Self-Management as an Intervention Following ePRO-Symptom Monitoring

Interventions following ePRO-symptom monitoring may include referrals, prescription of (co-)medication, switching or discontinuation of therapy, dose modification, or combinations of supporting electronic Health (eHealth) interventions and self-management recommendations [24,67]. All of these aim to improve patients' HRQOL and/or symptom self-management. Self-management is the individual's ability to manage symptoms, treatment, physical and psychosocial consequences, and lifestyle changes, inherent in living with a chronic condition. It comprises sufficient knowledge, adequate skills, and effective confidence to achieve attainable goals. The belief that one can successfully execute behavior required to produce an expected outcome in relation to consequences of cancer and its treatment is defined as self-efficacy. Self-efficacy is an important prerequisite of successful self-management. Self-management advice is not yet a common feature of ePRO-infrastructures for routine care [15]. As far as we know, OncoKompas is the only monitoring system that uses reasoning pathways to generate personalized self-management advice for cancer survivors based on ePRO data and cancer type [1]. Moreover, the eRAPID infrastructure includes self-management advice for mild or moderate symptoms and showed improved patient self-efficacy [25].

Self-management can be guided if health status is known, e.g., by monitoring self-management support (SMS) needs, self-efficacy to manage symptoms, symptom distress, and HRQOL [1]. Usually, SMS is an educational intervention that is part of an ongoing process to facilitate patient self-management behavior. Examples of SMS projects are Improving Patient Experience and Health Outcomes Collaborative (iPEHOC) [68], Symptom Management Improves your LifE [69], OncoKompas [67], and the Symptom Navi Program [70,71]. However, current evidence on SMS remains inconclusive to date [72]. A recent review of systematic reviews on post-treatment online interventions found only substantive evidence for the improvement of psychosocial and physical effects, particularly for improving fatigue and cognitive functioning [73]. There is a lack of SMS guidelines

and little clarity on what components work for improving outcomes. Best practices are obscured by varying terminology, incomplete description of the intervention approach, and heterogeneity of outcomes. Therefore, despite a growing body of research, it is difficult to draw firm conclusions regarding best practices guiding SMS components and approaches to delivering interventions, particularly in the field of eHealth-enhanced SMS in survivorship care [72].

Future Developments

The minimal use of SMS is a missed opportunity to engage patients in the use of ePRO data since they are largely responsible for managing most HRQOL-related issues in daily life, outside of clinic visits. Therefore, the Global Partners on Self-Management in Cancer have recently defined six key actions to improve and accelerate the use of SMS in survivorship care [72]. It not only requires SMS programs and infrastructures, but also changes in the active involvement of patients, in HCP communication and SMS skills, and in organization of health services.

*3.4. ePRO-Symptom Monitoring Implementation*

Focal points for implementation of ePROs in routine care also apply to implementation of ePRO-symptom monitoring. There are, however, several nuances. In comparison to general ePRO monitoring over longer time intervals, the often high-frequency reporting of ePRO-symptoms and the subsequent alerts are more time-consuming and intensive for patients and HCPs. Despite this time investment, patients show moderate to high compliance (51–92%) when using a monitoring application [22,33,73]. For HCPs, broad clinical adoption of symptom monitoring is limited particularly due to their doubts and resistance to a large volume of alerts [22]. In routine cancer care, especially ad hoc management is performed against escalating symptoms. Now alerts need to be handled and managed accordingly, requiring extensive adjustments in the clinical workflow. The impact on the workflow depends greatly on the active or reactive handling of alerts. In case of the reactive approach, patients are advised to contact the HCP, which will be less time consuming for HCPs, but may also be less effective [74]. Usually, nurses are primarily involved in handling alerts. Nurse-led management of symptom monitoring has shown to be feasible and effective [13,22,24].

Other reasons for limited adoption of ePRO-symptom monitoring are lack of expected or perceived value and a lack of broadly available oncology-specific symptom monitoring platforms (that are now rapidly emerging commercially) [49]. However, following proper implementation, nurses as well as oncologists, are highly compliant in reviewing the results and consider it overall useful [22,24].

To drive maximal adoption of ePRO-symptom monitoring for clinically meaningful use, the iPEHOC intervention has been developed. It aimed at implementation in multi-site (*n* = 6) Canadian oncology practices and was used for 6000 cancer patients [73]. Key factors for successful uptake were a supportive leadership structure that established ePROs use as a performance metric and building clinician capacity and confidence in interpreting and responding to ePRO-symptoms. Another key factor was obtaining extra information following high-severity symptom scores that prolonged over time. This extra information was an ePROM about the particular issue, which was automatically forwarded to the patient using algorithms in the online application. The iPEHOC implementation methods toolkit is available online [68].

An example for ePRO-symptom implementation and subsequent self-management is the CHEMO-SUPPORT approach [75]. The CHEMO-SUPPORT intervention applies motivational interviewing and goal-directed self-management coaching. With this coaching, ePRO-symptoms are embedded in the broader vision on nursing and SMS. Another example is the Symptom Navi Program, which describes implementation of a nurse-led intervention. It is especially developed to guide patients in self-management applying

the principles of SMS, based on behavior change theories [70,71]. The integration of the Symptom Navi Program as an eHealth intervention is currently investigated.

Instructions for the patient to act if severe or unexpected symptoms occur is extremely important to include in implementation plans, especially when ePRO-symptom monitoring is performed during the active treatment phase. In the case of these symptoms, patients need to follow the regular (offline) procedures and should in no case postpone reporting until the planned timing in the online application [21,33].

Future Developments

Alert algorithms for symptom monitoring will be continuously refined, using artificial intelligence approaches. We think international multidisciplinary collaboration is desirable, as it is a very time-consuming and delicate piece of work. This collaboration should include information technology engineers as well as machine learning and artificial intelligence specialists. Collaboration is also warranted for algorithms that provide guidance towards self-management because this feature turns the application into a medical aid and introduces additional legal requirements [76]. In all developments, minimization of the symptom alert burden on nurses, in addition to optimal safety for patients, is of utmost importance [22].

Development and implementation of the many applications on ePRO-symptom monitoring in survivorship care are costly. To our knowledge, one trial on symptom monitoring in follow-up for lung cancer showed it to be a cost-effective strategy [77]. This means that the medical profession considers the clinical evidence on effectiveness of these applications firm enough to continue broad implementation. In our view, however, cost evaluation is required, and cost-effectiveness models need to include questions about the value of care [35,78].

We strongly recommend that future developments and implementation strategies apply the principles of public and patient involvement, as recommended by the conceptual framework of the eHealth enhanced chronic care model [36]. Patient involvement should incorporate attention for patients with low health or eHealth literacy, mainly because they are reported to benefit the most. A trial evaluating ePRO-symptom monitoring in routine cancer care reported higher relevance and importance for communication with the HCP if patients had no prior experience with connected technologies and if they were less educated [22].

## 4. ePROs Ethical Considerations

While ePRO use in cancer control offers opportunities for improvement in cancer outcomes, it presents several ethical challenges that need to be recognized and addressed proactively. Many of these are not unique to (e)PROs, but rather inherent to the delivery of health care in general, such as considerations for privacy, confidentiality, and ownership of data. Systematic collection of any data requires balancing interests of an individual versus the population. If data collection only serves population-outcome monitoring, the individual may not derive any or only minimal personal benefit. For ePROs, this is the case if patients and HCPs do not receive feedback after the patient completes the questionnaires.

The important ethical considerations particularly relevant to ePRO use revolve around justice and fairness. As ePRO collection is a way of systematizing patients' perception of outcomes that are important to them, it is critical that the ePRO application is relevant, appropriate, acceptable, and accessible to all those whose perspectives it is meant to represent. However, patients who feel unwell, those with multiple chronic conditions or other complex needs, i.e., for whom ePRO collection could be particularly useful, may find it more cumbersome to provide data. However, it is precisely their voice that must be heard to provide the insights to what the unmet needs are [79]. In contrast, if self-management approaches provide support to patients who have moderate problems, then this can free limited health care resources to treat those most in need.

Furthermore, most ePRO data are collected in higher income countries [5], and therefore many patient groups are underrepresented. As a consequence, the rapid advancement of using technologies in health care raises the problem of the digital divide and subsequent data poverty [80]. Digital divide refers to the gap between demographics and regions that have access to modern information and communications technology and those that do not or have restricted access [81,82]. Data poverty is the inability for individuals, groups, or populations to benefit from discovery or innovation due to a scarcity of adequately representative data [80]. This includes the groups of patients with lower health or eHealth literacy and those with limited access to technology as they may not engage with ePRO collection [83,84]. It results in the fact that aggregated datasets used to develop and validate eHealth technologies might be safe and effective for some people, but less beneficial for others. Language barriers pose similar challenges.

Ensuring minimal burden of data collection and access to appropriate formats including translation of questionnaires, or use of images rather than text, is critical in facilitating access. It is also an advantage if an application is publicly available, without license and fee or royalty, such as PROMIS [11]. Another solution for equity and access to care is to provide patients without a smartphone with a device for ePRO monitoring, which has been successfully implemented in some US hospitals already [33].

Individuals and populations that are sources of data should not be just the passive recipients of the data capturing tools. They should rather be autonomous agents who understand the purpose of the data collection, see it as relevant and appropriate to their needs, and applied in the format that is acceptable to them. Without it, ePRO collection is just shifting responsibility for health care management onto a patient without any support. This again stresses the requirement of an early, proactive co-design process with patients before implementation of ePRO collection. Relevance and appropriateness also need to be embedded into the overall approach that supports SMS as an integral component of survivorship care [73]. An example of such successful experience-based co-design is a process of development of a PRO dataset for Indigenous Australians [85].

In addition, HCPs should use ePROs to optimize patient involvement. As patients are increasingly empowered to monitor and co-coordinate their disease, they need to be involved in the coordination and communication of care [72]. ePROs are one of the means to make this possible if they are actively used to feed the discussions with HCPs before making informed decisions. As such, the use of ePRO's appears to be promising to amplify the patients' voice in their own cancer care.

To address these challenges, implementation of ePROs in routine survivorship care requires collaborative consideration of its purpose, to ensure that the expected benefit of ePRO collection outweighs the risks and burdens of data collection and storage. This careful balance between beneficence and non-maleficence needs consideration at the individual patient level to provide appropriate support. It also needs consideration at the population level to ensure that the access to ePRO data is not biased and does not accentuate disparities [78].

## 5. Conclusions

Due to the narrative approach of this review, no systematic literature search or quality assessment was performed, introducing selection bias. Therefore, interpretation needs to be done with this in mind.

This review provides the reader with a comprehensive overview of many steps that have already been taken in the development, collection, and use of ePRO(-symptom) data for survivorship care. It showed the developments that are desirable for the future, especially for the routine and patient-centered use of ePROs. It is important to share, reuse, and incorporate the existing knowledge in future work, as many health care services and systems around the world are still in the starting phase of ePRO implementation. Regardless of the infrastructure that is used for ePRO collection, international developments can be unlocked by providing controlled access to ePRO data, algorithms, and models.

The use of ePROs is instrumental in driving innovation in the eHealth era. For symptom management, it is shifting care from reactive to proactive and preventive by integrating predictive devices into the patient's daily life [86]. All in all, patients and HPCs always have the choice about whether, in what way, and to what extent to utilize ePROs and subsequent self-management interventions. Currently, the range of choices is expanding rapidly. In contrast, ePRO monitoring provides an overview, thereby narrowing the scope as it comes to addressing unmet needs and delivering survivorship care.

**Author Contributions:** Authors C.J.G.v.d.H., F.M., M.E., R.J.C., A.B., G.G., I.W., A.C. (Annemarie Coolbrandt), M.L., G.V., A.C. (Andreas Charalambous), B.K., D.H., E.M.B. and L.V.v.d.P.-F. have contributed to the conceptualization, review and editing of the manuscript; Original draft preparation, C.J.G.v.d.H., F.M., M.E., R.J.C., A.C. (Andreas Charalambous), B.K., D.H. and L.V.v.d.P.-F.; visualization, C.J.G.v.d.H.; All authors have read and agreed to the published version of the manuscript.

**Funding:** R.J.C. received salary support from National Health and Medical Research Council (APP1194051). The remaining authors received no external funding.

**Conflicts of Interest:** Author C.J.G.v.d.H. has received institutional implementation and research grants from AstraZeneca, Boehringer-Ingelheim, Bristol Myers Squibb, Ipsen and Merck. Author M.E. received institutional research grants from Kaiku Health, Bristol Myers Squibb and Roche and institutional fees as a Scientific Advisory Board Member/Consultant from Roche. Author G.V. reports personal fees from Roche, Eisai, Novartis, Seattle Genetics and Sanofi, and institutional grants from Breast Cancer Now, EORTC, Yorkshire Cancer Research, Pfizer and IQVIA, outside the submitted work. Author E.M.B. receives personal fees as a scientific advisor/consultant to the following entities that provide bio-medical services and could give the appearance of potentially influencing the work: AstraZeneca, Carevive Systems, Navigating Cancer, Resilience, N-Power, and Sivan Healthcare. Author D.H. receives personal fees as a Scientific Advisory Board Member/Consultant to Carevive Systems and institutional grants from Astra Zeneca. Author A.C.(Andreas Charalambous), has received speaker honoraria and consultant fees from AstraZeneca, Daichy Sankyo, SPCC, MSD and project grants from Takeda and Amgen. Author R.J.C. has received speaker honoraria from Dr Reddy's Laboratory and Stratpharma. The remaining authors declare no conflicts of interest.

## Abbreviations

| | |
|---|---|
| AE | Adverse Event |
| ASyMS | Advanced Symptom Monitoring and Management System |
| CTC-AE | Common Terminology Criteria for Adverse Events |
| eHealth | electronic Health |
| HCP | Health Care Practitioners |
| HER | Electronic Health Record |
| EORTC | European Organisation for Research and Treatment of Cancer |
| ePRO | electronic Patient-Reported Outcome |
| ESAS | Edmonton Symptom Assessment Scale/System |
| FDA | Food and Drug Administration |
| GP | General Practitioner |
| HCP | Health Care Practitioner |
| ICHOM | International Consortium for Health Outcomes Measurement |
| iPEHOC | Improving Patient Experience and Health Outcomes Collaborative |
| ISOQOL | International Society for Quality of Life Research |
| MSAS | Memorial Symptom Assessment Scale |
| PHR | Personal Health Record |
| PRO | Patient-reported Outcome |
| PRO-CTC-AE | Patient-Reported Outcomes Common Terminology Criteria for Adverse Events |
| PROM | Patient-Reported Outcome Measure |
| PROMIS | Patient-Reported Outcomes Measurement Information System |
| PROFILES | Patient-Reported Outcomes Following Initial treatment and Long term Evaluation of Survivorship |

| SISAQOL | Setting International Standards in Analyzing Patient-Reported Outcomes and Quality of Life Endpoints Data |
| SMS | Self-Management Support |
| VBHC | Value Based Health Care |

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
