# Peer review of "A Narrative Review on the Collection and Use of Electronic Patient-Reported Outcomes in Cancer Survivorship Care with Emphasis on Symptom Monitoring"

_curroncol, doi:10.3390/curroncol29060349_

Round 1

Reviewer 1 Report

Thank you for the opportunity to review this paper on the topic of collection and use of electronic patient reported outcomes in cancer survivorship care with emphasis on symptom monitoring. This is a well written narrative review. The majority of my comments are minor in nature and I have detailed them below.

One substantive comment for the authors, relates to the populations for whom electronic patient reported outcomes in cancer survivorship are tailored to. At no point is there a discussion related to specific populations whose needs may be different than the ‘average patient’ (based on my read of this paper, the average paper is a well-educated and technologically capable middle-aged person (not too old or too young) who manages their own health). While there is some discussion about data poverty and the digital divide, there isn't consideration of how electronic patient reported symptom monitoring is relevant to specific populations, i.e. pediatrics, young adults, or geriatrics. Given the comprehensive nature of this article, I would consider including it.  

Given the extensive nature of the article, I would also consider including some figures to illustrate the main issues ‘at a glance’.

Abstract

Cancer is not mentioned in the abstract. It should be clear that this is a narrative review about ePROs in Cancer survivorship care.

Background

Again, first sentence of the paper should emphasize that this is a narrative review about ePROs in Cancer survivorship care.

Could the authors please break up the following sentence: “The narrative methodology is used to obtain a broad perspective and we did not intent to locate all relevant literature.”- And clarify that this was not a systematic search of the literature?

Section 1- ePRO methodology

Line 130, reads: “Then the patient reports several times a day for a few weeks, the so called Ecological Momentary Assessment [29,30]” This is an incomplete sentence.  

In line 135, can the authors please explain how ePROs can speak to financial toxicity?

In line 139, the transition from one sentence to the next is awkward. Please revise.

Line 140-145 describes some of the aspirational ideas of ePROs as being shared in the MDT- it would be helpful to share some of the known challenges/barriers to doing this.

Section 1.2

Line 175 starts with “besides”- which should be deleted for an alternative transition word (happens again on line 191- suggest to search and replace this word- same goes for the term “on the other hand”).

Section 2- ePRO monitoring

Self-management seems to be a big part of the article – it is one of the keywords- yet it is not defined until line 321. It would be helpful to define this term for readers early on. What is it and why is it important?

There is no section 3?

Section 4- Ethical issues

Please provide a reference to support the definition of digital divide on line 448

Conflicts of Interest

Please ensure that all authors have disclosed all relevant conflicts of interest.

I hope these comments are helpful and wish you luck with your revision.

Author Response

Thank you for the opportunity to review this paper on the topic of collection and use of electronic patient reported outcomes in cancer survivorship care with emphasis on symptom monitoring. This is a well written narrative review. The majority of my comments are minor in nature and I have detailed them below.

We thank the reviewer for his/her detailed comments and feedback.

One substantive comment for the authors, relates to the populations for whom electronic patient reported outcomes in cancer survivorship are tailored to. At no point is there a discussion related to specific populations whose needs may be different than the ‘average patient’ (based on my read of this paper, the average paper is a well-educated and technologically capable middle-aged person (not too old or too young) who manages their own health). While there is some discussion about data poverty and the digital divide, there isn't consideration of how electronic patient reported symptom monitoring is relevant to specific populations, i.e. pediatrics, young adults, or geriatrics. Given the comprehensive nature of this article, I would consider including it.  

This is a very valuable remark. Thank you. We have added a few sentences on this topic in section 2.2.

Given the extensive nature of the article, I would also consider including some figures to illustrate the main issues ‘at a glance’.

Thank you for this advice. We have added a graphical abstract to the paper as is included as an option in the author guidelines for Current Oncology.

Abstract

Cancer is not mentioned in the abstract. It should be clear that this is a narrative review about ePROs in Cancer survivorship care.

We thank the reviewer for this detailed comment. We have adapted it accordingly in line 39.  

Background

Again, first sentence of the paper should emphasize that this is a narrative review about ePROs in Cancer survivorship care.

We have added this indeed (in line 55).

Could the authors please break up the following sentence: “The narrative methodology is used to obtain a broad perspective and we did not intent to locate all relevant literature.”- And clarify that this was not a systematic search of the literature?

We have added this remark, thereby clarifying that we did not undertake a systematic search (in line 65). 

Section 1- ePRO methodology

Line 130, reads: “Then the patient reports several times a day for a few weeks, the so called Ecological Momentary Assessment [29,30]” This is an incomplete sentence.  

We have changed the sentence into: “In this so-called Ecological Momentary Assessment, the patient reports on his or her problems several times a day for a few weeks.”

In line 135, can the authors please explain how ePROs can speak to financial toxicity?

We have clarified this sentence by changing “using ePROs to capture financial toxicity” into “using ePROs to screen for financial toxicity”.

In line 139, the transition from one sentence to the next is awkward. Please revise.

We have adapted the second sentence into: “In order to provide personalized survivorship care, it is helpful…”

Line 140-145 describes some of the aspirational ideas of ePROs as being shared in the MDT- it would be helpful to share some of the known challenges/barriers to doing this.

We thank the reviewer for this suggestion. We have added some challenges accordingly in lines 154-156.

Section 1.2

Line 175 starts with “besides”- which should be deleted for an alternative transition word (happens again on line 191- suggest to search and replace this word- same goes for the term “on the other hand”).

In the manuscript the word ‘besides’ was used 3 times (lines 181, 198, 219) and we replaced them, as well as the 2 times we used ‘on the other hand’ (lines 456, 517). 

Section 2- ePRO monitoring

Self-management seems to be a big part of the article – it is one of the keywords- yet it is not defined until line 321. It would be helpful to define this term for readers early on. What is it and why is it important?

We thank the reviewer for this suggestion, we have added the importance of self-management in section 1.1.

There is no section 3?

We thank the reviewer for detecting this detailed mistake in the paper. 

Section 4- Ethical issues

Please provide a reference to support the definition of digital divide on line 448

We have added the references (line 464).

Conflicts of Interest

Please ensure that all authors have disclosed all relevant conflicts of interest.

All authors have updated the COIs accordingly.

I hope these comments are helpful and wish you luck with your revision.

Thank you.

Reviewer 2 Report

This is a narrative review about the "Collection and Use of Electronic Patient-Reported Outcomes in Cancer Survivorship Care with Emphasis on Symptom Monitoring". As a narrative review, it is not intended to be comprehensive, systematic or quantitative. However, it is very well written and it is logical i.e. presents all important aspects (e.g.selection of questionnaire, implementation, monitoring etc) with paragraph about future development for each section.

Author Response

We thank the reviewer for the positive feedback.